# Vibration Perception Thresholds of Skin Mechanoreceptors Are Influenced by Different Contact Forces

**DOI:** 10.3390/jcm10143083

**Published:** 2021-07-13

**Authors:** Claudio Zippenfennig, Bert Wynands, Thomas L. Milani

**Affiliations:** Department of Human Locomotion, Faculty of Behavioral and Social Sciences, Institute of Human Movement Science and Health, Chemnitz University of Technology, 09107 Chemnitz, Germany; bert.wynands@hsw.tu-chemnitz.de (B.W.); thomas.milani@hsw.tu-chemnitz.de (T.L.M.)

**Keywords:** vibration perception threshold, mechanoreceptors, sensory perception, spatial summation, Meissner and Pacinian corpuscles

## Abstract

Determining vibration perception thresholds (VPT) is a central concern of clinical research and science to assess the somatosensory capacity of humans. The response of different mechanoreceptors to an increasing contact force has rarely been studied. We hypothesize that increasing contact force leads to a decrease in VPTs of fast-adapting mechanoreceptors in the sole of the human foot. VPTs of 10 healthy subjects were measured at 30 Hz and 200 Hz at the heel of the right foot using a vibration exciter. Contact forces were adjusted precisely between 0.3 N–9.6 N through an integrated force sensor. Significant main effects were found for frequency and contact force. Furthermore, there was a significant interaction for frequency and contact force, meaning that the influence of an increasing contact force was more obvious for the 30 Hz condition. We presume that the principles of contrast enhancement and spatial summation are valid in Meissner and Pacinian corpuscles, respectively. In addition to spatial summation, we presume an effect on Pacinian corpuscles due to their presence in the periosteum or interosseous membrane.

## 1. Introduction

Mechanoreceptors in the skin of human feet provide important information about various environmental stimuli [1]. If this feedback system fails, e.g., due to diseases such as diabetes mellitus or Parkinson’s disease, gait and balance often deteriorate [2,3]. It is therefore a central concern of clinical research and science to assess the somatosensory capacity of humans, as a decrease in sensitivity may be an early sign of peripheral neuropathy [4].

Four different types of cutaneous mechanoreceptors are responsible for sensing and transmitting diverse pressures and vibrations at the sole of the foot: slowly adapting (SA) type I (Merkel discs) and II (Ruffini corpuscles) receptors, which react to continuous pressure, and fast-adapting (FA) type I (Meissner corpuscles) and II (Pacinian corpuscles) receptors, which respond to the onset and offset of stimulation. SA I and FA I receptors are located near the skin surface and have small and well-defined receptive fields. SA II and FA II receptors lie deeper in the skin and have large receptive fields with obscure boundaries [5,6]. In this study, we focused on Meissner (FA I) and Pacinian (FA II) corpuscles.

Determining vibration perception thresholds (VPT) is a form of quantitative sensory testing. To produce reliable results, it is essential to standardize procedures. Numerous influencing factors for determining VPTs have already been studied. These include frequency [6,7,8], anatomical location [8], size of the probe [9,10], mechanical skin properties [1,11], temperature [12], gender [13], and age [14]. One parameter that has received little attention to date is the quantity of the load (pressure, acceleration, and contact area) with which the indenting object stresses the corresponding area of the subject’s skin.

Over the last 15 years, research studies measuring VPTs have used different contact forces, ranging from no force offset to avoid pre-activation of mechanoreceptors before the vibration stimulus [8], a preload of 2 N [11], and a preload of 4 N to simulate the pressure under the foot during walking [15]. These different contact forces may restrict the comparison of VPTs, since different studies have shown that increasing preloads from 0.1 N up to 8 N leads to an improvement in sensitivity [16,17,18]. In contrast, Hagander et al. [19] found decreased sensitivity at preloads of 1 N compared to 0.3 N and 0.5 N at the index finger. At the big toe, increasing preloads had no influence on VPTs [19]. Furthermore, Gregg [20] and Era and Hänninen [21] confirmed the results of Hagander et al. [19] at the big toe, finding no dependence between different contact forces and VPT. Two recently published studies evaluated the influence of standing compared to sitting while measuring VPTs at the sole of the foot [22,23]. Interestingly, these two studies also had contradictory results. While Mildren et al. [22] partially found a significant increase of VPT while standing in comparison to sitting, Germano et al. [23] found no significant differences. As both studies showed higher contact forces when standing compared to sitting, these two studies reflect the contradictions in the current literature.

To our knowledge, only Gu and Griffin [24] have examined the influence of different contact forces on VPT at the sole of the foot [24]. Their results exhibit slight VPT decreases with increasing force at the hallux and the ball of the foot. VPT decreases were more pronounced at 160 Hz than at 20 Hz [24]. In contrast to Gu and Griffin [24], our study did not use a static probe surrounding, which would compress the skin around the probe as well. Instead, only the vibrating probe exerted the contact force. A wide range of contact forces (0.3–9.6 N) was examined, exceeding typically-used values (0–4 N) in previous studies [8,11,15], to detect possible ceiling effects. To evaluate the effect on Meissner and Pacinian corpuscles separately, VPTs were collected at the most sensitive frequency of each receptor type. We hypothesized that increasing contact forces leads to a decrease in VPTs at both mechanoreceptors, which results in improved foot sensitivity. Finally, we tested the hypothesis that there is an interaction between sensor type and increasing contact forces.

## 2. Materials and Methods

Ten healthy young subjects participated in this study (8 ♂/2 ♀; mean ± SD: 22.9 ± 1.4 years, 74.9 ± 9.7 kg, 178.8 ± 9.7 cm). All participants gave their written consent and were free of any diseases that could affect the sensory system (diabetes mellitus, peripheral neuropathy, neurological diseases, etc.). All procedures were performed in accordance with the recommendations of the Declaration of Helsinki. The measurement system and methodology were approved by the Ethics Committee of the Faculty of Behavioural and Social Sciences of the corresponding university for a similar study in our work group (V-277-17-DS-KUS/WUS-22062018) [25]. The sensory measuring procedure has been validated and used in several published studies [1,23,26].

A modified vibration exciter (Typ 4180, Brüel & Kjaer Vibro GmbH, Darmstadt, Germany) powered by a powerbank (XTPower MP-3200, Batteries and Power Solutions GmbH, Ellwangen, Germany) was used to measure VPTs (Figure 1). To exclude mechanical influencing factors, the vibration amplitude (µm) was calculated using an external acceleration sensor (MMA2240KEG, NXP Semiconductors Netherlands B.V., Eindhoven, The Netherlands) placed in series with the probe of the vibration exciter (Figure 1). To measure the vibration amplitude as accurately as possible, the vertical movement of the vibration exciter’s probe (diameter 7.8 mm) was calibrated before the measurements using a high-precision capacitive position sensor (CS05, Micro-Epsilon Messtechnik GmbH & Co. KG, Ortenburg, Germany). This ensured direct readings of the vibration amplitude during VPT measurements. Using a swivel arm, the probe of the vibration exciter was placed precisely perpendicularly at the plantar heel. A force sensor (DS050A9, disynet GmbH, Brüggen-Bracht, Germany), also placed in series with the probe of the vibration exciter (Figure 1), was used to adjust different contact forces: 0.3 N, 0.6 N, 1.2 N, 2.4 N, 4.8 N, and 9.6 N. This was done to cover an extensive range of contact forces. Previous pilot measurements showed that approximately 10 N was the upper limit, both from a practical point of view and for the wellbeing of the subjects. The contact force was continuously monitored and, if necessary, readjusted between individual measurements. The investigation started with the lowest contact force of 0.3 N. To give the skin sufficient recovery time and to prevent accumulating effects, the resting times between the individual contact forces were also increased exponentially (from 30 s to 8 min) to accommodate possible disadvantages of non-randomization.

VPTs were measured at the heel of the right foot at two different frequencies: 30 Hz and 200 Hz. These two frequencies are considered ideal for measuring the vibration perception of Meissner and Pacinian corpuscles [6,7]. After a 10-min acclimation period, participants lay supine with their right foot fixed in a cushioned splint (Figure 1). VPTs were measured three times for each contact force using a self-written binary search method [15,22]. Participants had to press a button as soon as they felt a vibration. Starting with a sinusoidal vibration burst above the participants’ individual thresholds (2 s duration followed by a 2–7 s pause), the program reduced or increased the stimulus intensity automatically, depending on whether the subject felt the last stimulus. The algorithm stopped four bursts after the first undetected stimulus. The mean of the last recognized and the last unperceived vibration stimuli was determined as VPT. A graphical visualization of the measurement algorithm can be found in Appendix A (Figure A1). If the subjects pressed the button more than twice while no stimulus was present, the measurement was repeated. Out of a total of 360 measurements, only four had to be repeated (~1%). A single VPT measurement took about 2 min. The three measurements per contact force thus resulted in a duration of approximately 6 min. The skin recovery time lasted a total of 15 min and 30 s. Including acclimatization, test trials, and storing the VPTs, this resulted in a total duration of approximately one and a half hours per test subject. Therefore, the two different frequencies were measured on two different days to allow subjects to concentrate for as long as possible. Since two different mechanoreceptors were addressed by the selected frequencies, learning effects were not expected.

Participants wore noise-cancelling headphones (QuietComfort 25 Acoustic Noise Cancelling headphones, Bose Corp., Framingham, MA, USA) to eliminate environmental noises. Temperature values of the foot were monitored using a non-contact infrared thermometer (UNI-T UT301C, Batronix GmbH & Co. KG, Preetz, Germany).

The mean out of three VPT measurements per contact force condition was used for statistical analysis with R [27]. Data was checked for outliers, and the benefit of logarithmical transformation due to non-normal distribution was tested. After logarithmic transformation, the distribution of the data was still non-normal. The Aligned Rank Transform (ART) procedure was used to examine interaction effects [28]. A post hoc analysis was performed using interaction contrasts, looking at differences of differences [28].

## 3. Results

Plantar temperature was within acceptable ranges (30 Hz pre vs. post: 22.1 ± 1.5 vs. 23.6 ± 2.7; 200 Hz pre vs. post: 23.4 ± 2.9 vs. 24.2 ± 2.4) [12]. VPTs decreased at both frequencies with increasing contact forces (Table 1, Figure 2 and Figure 3).

A significant main effect was found for ‘frequency’, indicating lower thresholds (higher sensitivity) for 200 Hz measurements (F(1,99) = 306.837, *p* < 0.001, η_p_^2^ = 0.76). Furthermore, there was a significant main effect for ‘contact force’, meaning lower thresholds (higher sensitivity) with increased contact forces (F(5,99) = 19.610, *p* < 0.001, η_p_^2^ = 0.50). Additionally, a significant interaction effect was observed for ‘frequency’ and ‘contact force’, which means that increased contact force led to higher reductions in VPTs at 30 Hz compared to 200 Hz (F(5,99) = 12.947, *p* < 0.001, η_p_^2^ = 0.40).

Interaction contrasts (Figure 4) were found for the difference between VPTs at 200 Hz and 30 Hz in the contact force condition 0.3 N compared to the differences at 2.4 N, 4.8 N, and 9.6 N (all *p* < 0.001). Further significant differences of differences were found between 0.6 N and 2.4 N (*p* = 0.018), 4.8 N, and 9.6 N (both *p* < 0.001) and between 1.2 N and 4.8 N (*p* = 0.005) and 9.6 N (*p* = 0.001).

## 4. Discussion

The aim of the present study was to investigate the effects of increasing contact forces on VPTs of two different mechanoreceptors (Meissner and Pacinian corpuscles) in healthy young subjects. We hypothesized that greater contact forces lead to lower VPTs at both mechanoreceptors and, based on their anatomical location, that there is an interaction between sensor type and increasing contact force.

In accordance with the literature, we found a significant main effect for ‘frequency’, indicating lower thresholds for 200 Hz compared to 30 Hz [8,22]. Furthermore, we found a significant main effect for ‘contact force’, meaning that an increased contact force led to lower VPTs and, therefore, to better sensitivity. Furthermore, we found a significant interacting effect between ‘frequency’ and ‘contact force’, meaning that the influence of increasing contact force was different for Meissner than for Pacinian corpuscles. Figure 2 and Figure 3 show a more pronounced decrease at 30 Hz than at 200 Hz. Pairwise comparisons exhibited significant differences between lower (0.3 N–1.2 N) and higher contact forces (2.4 N–9.6 N) (Figure 4).

### 4.1. Meissner Corpuscles

Meissner corpuscles have circular receptive fields with several zones of maximal sensitivity [6]. This area of high sensitivity lies in the middle of the circular receptive field. Towards the periphery, the Meissner corpuscle is increasingly less sensitive [6]. The increasing pressure caused increased skin deformation, because the area of skin surrounding the probe also increased [29]. This may not only have led to a reduction in the distance between the stimulus and the perception hotspots, but also to a general increase in the number of Meissner corpuscles being addressed [29]. Additionally, increasing contact forces led to mechanical deformation of the non-neuronal components. In the Meissner corpuscles, force transmission through collagen fibers results in bending of the axon terminals [5]. This compression generates action potentials during stimulus onset [5]. High contact forces might have pre-bent axon terminals before the actual stimulus. Thus, it is possible that the low-frequency vibrations can be transmitted more directly to the axon and, thus, trigger action potentials more easily.

Furthermore, in the hallux, for example, Meissner corpuscles reach a density of approximately 10/mm^2^ [30]. Our probe stimulated a skin area of ~48 mm^2^. However, more surrounding skin was indented by the probe at higher pressures, which might have increased the affected area. Furthermore, skin mechanics change due to compression. The compressed skin may transmit vibrations farther across the skin [1]. Thus, more Meissner corpuscles can be used for stimulus detection. Interestingly, Gu and Griffin [10] challenge the theory of the increasing number of responding Meissner corpuscles [10]. They tested the spatial summation for vibrotactile stimuli of Meissner and Pacinian corpuscles at the foot. They used different probe sizes (1–10 mm in diameter) and vibration frequencies of 20 Hz and 160 Hz. They showed that the VPT decreased with increasing area only at a frequency of 160 Hz (representative for Pacinian corpuscles) [10]. Therefore, spatial summation only seems to occur at higher frequencies [9,10].

In addition to our approach regarding perception hotspots described above, the synergistic cooperation of slowly adapting type I receptors (Merkel discs) and rapidly adapting receptors (Meissner corpuscles) may provide a second explanatory model. Meissner corpuscles are relatively insensitive to static forces and have a high sensitivity at low spatial resolutions [31]. On the other hand, Merkel discs have a low sensitivity at high spatial resolutions [31]. According to Abraira and Ginty [31], it is conceivable that Merkel discs and Meissner corpuscles act together to encode a more complete picture of tactile space [31]. Merkel discs react to mechanical forces on the skin with a persistent and graded dynamic, followed by ‘bursting’ at irregular intervals, which correlates linearly with penetration depth [31,32]. We know that Merkel disc receptors also encode low frequency vibrations (5–15 Hz) and that the transition between the frequency coding of the receptors overlaps [33]. Therefore, the increasing contact force could have led to an increased synergistic response behavior of Merkel discs. This synergistic interaction could have led to the increased sensitivity measured during the 30 Hz condition.

Based on this synergistic interaction of these two mechanoreceptors, the increased sensitivity with increasing contact forces may also be attributed to contrast enhancement. Due to the receptor density and the overlapping receptive fields, sensory information is mediated simultaneously by several receptors [34]. As a result, the functional organization of the sensory processing networks is hierarchical [34]. Laterally inhibiting interneurons reduce or inhibit the excitation of weak signals at the receptors’ first interconnection level due to the activity in the neurons with strong signals [35]. As already mentioned, more receptors may be involved in stimulus detection with increasing contact force. To enable the neural network to precisely identify the stimulus source, i.e., the probe, adjacent weaker stimuli stemming from skin deformation may be more severely inhibited than for lower contact forces. At the same time, the signal of the strongly activated receptors directly under the probe would be amplified even more, so that the original stimulus (the vibration) could be perceived better.

### 4.2. Pacinian Corpuscles 

As already mentioned, the improvement in sensitivity of the Pacinian corpuscles with increasing contact force seems to be the result of spatial summation [10]. While Gu and Griffin [10] used different probe sizes and vibration frequencies, Kekoni et al. [9] described another point which may have led to spatial summation in our study. They used the same vibration exciter with a similar measurement methodology [9]. By directly applying the probe of the vibration exciter to the measurement site, the vibration stimulus can spread to surrounding skin areas [9]. Supported by the possibility of changing skin characteristics from increasing compression and the high sensitivity of the Pacinian corpuscles, these residual vibrations, depending on their intensity, can also cause reactions in surrounding Pacinian corpuscles [36]. The altered and far-reaching proliferation of vibrations with altered contact forces of the probe represents a possible reason for the improving sensitivity of the Pacinian corpuscles. The smallest residual vibration may already lead to the excitation of surrounding Pacinian corpuscles. This is supported by the assumption that mechanical skin properties and the reaction of the Pacinian corpuscles may act together due to the skin’s own resonance and the optimal sensitivity of the Pacinian corpuscles [36].

Additionally, Pacinian corpuscles are not only located in the dermis, but also in the periosteum or interosseous membrane [32,37]. As the contact force increases, the heel fat pad is compressed, and the probe and calcaneus come closer together. The vibration stimulus may therefore be transmitted to the Pacinian corpuscles located in the periosteum/interosseous membrane and, thus, possibly have been the reason for the improved sensitivity of the Pacinian corpuscles.

### 4.3. General Aspects and Limitations 

Many of our discursive approaches were interpretive, based on the interaction between mechanoreceptors, stimulus conduction, and stimulus processing. Unfortunately, we were limited by our technical possibilities and could only measure the outcome of mechanoreceptor stimulation in terms of VPTs. Using microneurography, including stimulus conduction and EEG-measurements, may provide information on stimulus processing. Furthermore, generalization of our discursive approaches is limited due to the small number of subjects, only one measured anatomical location, and one probe size. Based on a different receptor distribution and differences in the mechanical skin properties [1,38], an inclusion of further anatomical locations on the sole of the foot (e.g., first metatarsal head or big toe) and different probe sizes would be useful in further studies. An increase in the number of subjects is also conceivable. However, in sensory measurements, the neuronal responses that arise from stimulating receptors can be influenced by attention, cognitive processes, and other behavior patterns of the perceiver [39]. The success of sensory measurements depends to a large extent on the cooperation of the subject. To maintain high levels of concentration, the breaks described in the methods section for the skin to recover were also used as time for subjects to relax. In addition, the measurements at the two different frequencies were distributed over two days to ensure the active participation of the subjects at all times.

Based on our results, we were able to establish a relationship between VPTs and contact forces considering different measurement frequencies. If we disregard the two highest contact forces tested in this study (4.8 N and 9.6 N), the lowest variability for determining VPTs was seen at 1.2 N (Figure 4). The highest contact forces led to strong skin deformations, which could influence the response behavior of the mechanoreceptors and were unpleasant for the test subjects.

## Figures and Tables

**Figure 1 jcm-10-03083-f001:**
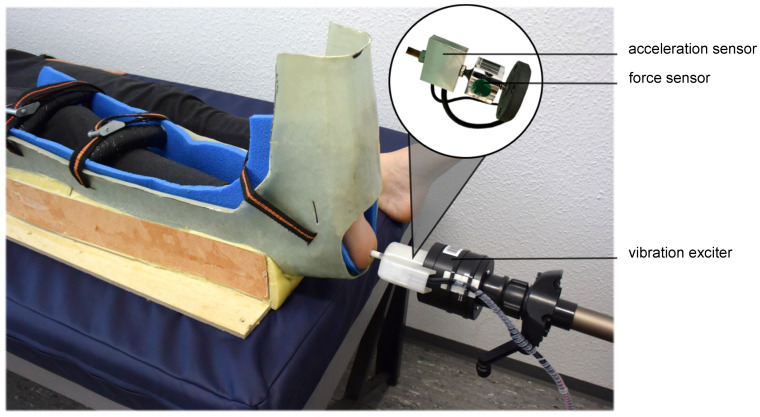
Measurement setup. The 7.8 mm diameter probe (area ~48 mm^2^) of the modified vibration exciter placed perpendicularly at the plantar heel of the right foot. To keep the test person’s foot as still as possible, the foot was strapped into a cushioned splint. The contact forces (0.3 N, 0.6 N, 1.2 N, 2.4 N, 4.8 N, and 9.6 N) were adjusted and continuously monitored using a force sensor. The amplitude of the vibration was measured directly in micrometers using an accelerometer.

**Figure 2 jcm-10-03083-f002:**
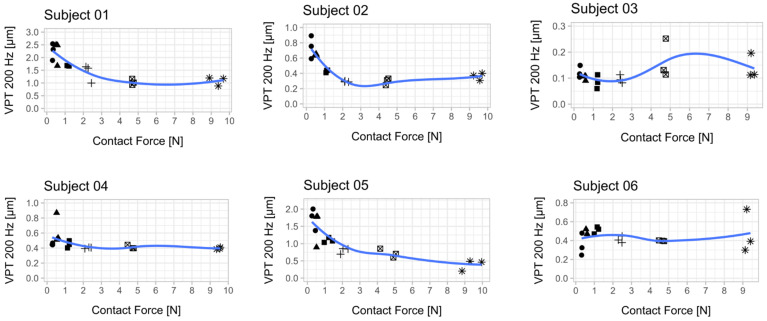
Relationship between vibration perception thresholds (VPT) at 200 Hz and contact force for each subject. Each scatter plot shows the three determined VPTs for each contact force condition. A smoothed conditional mean was added using the R function “geom_smooth ()” in combination with the argument “method = loess”. Seven out of 10 subjects showed a reduction in VPT with increasing contact force. The mean from the respective three VPT measurements per contact force was used for statistical analysis (Figure 4). An attempt was made to establish a psychophysical relationship between the contact force and the perceived sensitivity across all subjects (Appendix B, Figure A2).

**Figure 3 jcm-10-03083-f003:**
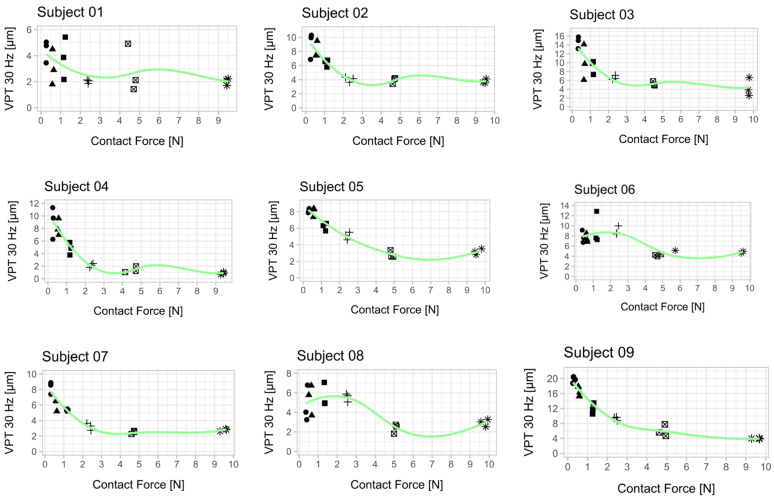
Relationship between vibration perception thresholds (VPT) at 30 Hz and contact force for each subject. Each scatter plot shows the three determined VPTs for each contact force condition. A smoothed conditional mean was added using the R function “geom_smooth ()” in combination with the argument “method = loess”. All 10 subjects showed a reduction in VPT with increasing contact force. The mean from the respective three VPT measurements per contact force was used for statistical analysis (Figure 4). An attempt was made to establish a psychophysical relationship between the contact force and the perceived sensitivity across all subjects (Appendix B, Figure A2).

**Figure 4 jcm-10-03083-f004:**
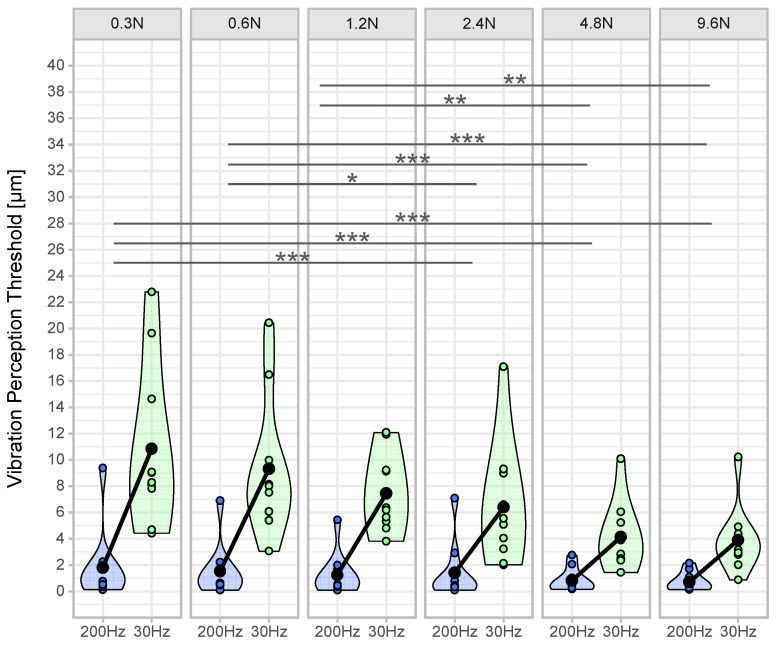
Vibration perception thresholds per contact force and frequency. The violin plots show the density of the distribution for the vibration perception thresholds (VPT) for the frequencies 200 Hz (blue) and 30 Hz (green) of the respective contact forces based on the width of the plot (*n* = 10 per contact force condition). The individual points within the violin plots represent the averaged VPT (mean out of three VPT measurements per contact force condition) of each subject. The black dot is the mean value of all subjects. The connecting line between the black dots represents the mean difference. Superscripted symbols represent significant interaction contrasts and differences of the differences (*** *p* < 0.001, ** *p* < 0.01, * *p* < 0.05).

**Table 1 jcm-10-03083-t001:** Vibration perception thresholds [µm] for each contact force (N) and frequency (Hz).

*n* = 10	0.3 N	0.6 N	1.2 N	2.4 N	4.8 N	9.6 N
VPT 30 Hz (µm)	10.9 ± 6.2	9.3 ± 5.3	7.5 ± 3.0	6.4 ± 4.5	4.1 ± 2.5	3.9 ± 2.5
VPT 200 Hz (µm)	1.8 ± 2.8	1.5 ± 2.0	1.2 ± 1.6	1.4 ± 2.2	0.8 ± 0.9	0.7 ± 0.7

Mean ± SD of vibration perception thresholds (VPTs).

## Data Availability

The data presented in this study are available on request from the corresponding author. The data are not publicly available due to further analyses.

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
