# Peer review of "Vibration Perception Thresholds of Skin Mechanoreceptors Are Influenced by Different Contact Forces"

_jcm, 2021, doi:10.3390/jcm10143083_

Round 1

Reviewer 1 Report

The authors investigated how vibration perception threshold is affected by frequency of vibration and the contact force of the applied vibration. They utilize a series of force and accelerometer sensors to measure contact force and vibration perception threshold on the sole of the foot in healthy participants. The vibrations targeted different mechanoreceptors based on the frequency of the vibration (30 or 200 Hz). The results showed a significant interaction effect of frequency and contact force, showing that an increase in force was related to a different change in perception threshold across vibration frequency, with greater improvements in perception for 30 Hz vibrations. The authors conclude/recommend that future studied should monitor and consider contact force when determining or measuring vibration perception. 

Revisions:

  1. Please add power/effect size of the statistical analysis, as the subject sample was relatively small. Showing good power/effect size will make the results much stronger in light of the smaller sample size.
  2. Please describe in the methods how exactly the algorithm changed the vibration intensity and what comprises the vibration intensity. Is it a change in vibration amplitude since the frequency is kept the same? This should be clarified. Perhaps a general range of amplitudes used should also be added.
  3. Please proof read the manuscript, I found several grammatical error. 

Author Response

Dear Reviewer 1,

thank you very much for your comments. Please find attached a point-by-point response to your comments.

Kind regards,

Claudio Zippenfennig

Reviewer 2 Report

This paper presents a significant and interesting research that highlights an important factor influencing the vibration perception thresholds (VPT). Through described research we would be able to measure the VPT more accurately.

However, the description of the methods and the quality of presentation could be slightly improved:

Line 122 – Why the known and described in literature psychophysical procedures, e.g. the adaptation method, were not used? Please justify the choice of psychophysics method. Why the described method has been chosen (please explain chosen termination criteria and threshold estimate)?

Line 135 – In what foot temperature range the tests were performed? What was done with the rising or falling foot temperature?

Table 1, Figures 2-4 – Wouldn't it be more advantageous to present the results on a logarithmic scale, e.g. displacement level or acceleration level, referring to Weber's law?

Line 285 – The conclusion that the contact force has to be controlled is rather obvious (ISO 13091-1). The authors' success is in determining the relationship between VPT and contact forces.

Author Response

Dear Reviewer 2,

thank you very much for your comments. Please find attached a point-by-point response to your comments.

Kind regards,

Claudio Zippenfennig
